# Functional Transcription Factor Target Networks Illuminate Control of Epithelial Remodelling

**DOI:** 10.3390/cancers12102823

**Published:** 2020-09-30

**Authors:** Ian M. Overton, Andrew H. Sims, Jeremy A. Owen, Bret S. E. Heale, Matthew J. Ford, Alexander L. R. Lubbock, Erola Pairo-Castineira, Abdelkader Essafi

**Affiliations:** 1MRC Institute of Genetics and Molecular Medicine, University of Edinburgh, Edinburgh EH4 2XU, UK; andrew.sims@ed.ac.uk (A.H.S.); bheale@gmail.com (B.S.E.H.); matthew.ford@mail.mcgill.ca (M.J.F.); alex.lubbock@vanderbilt.edu (A.L.R.L.); erola.pairo-castineira@igmm.ed.ac.uk (E.P.-C.); a.essafi@bristol.ac.uk (A.E.); 2Department of Systems Biology, Harvard University, Boston, MA 02115, USA; jaowen@mit.edu; 3Centre for Synthetic and Systems Biology (SynthSys), University of Edinburgh, Edinburgh EH9 3BF, UK; 4Patrick G Johnston Centre for Cancer Research, Queen’s University Belfast, Belfast BT9 7AE, UK; 5Department of Physics, Massachusetts Institute of Technology, Cambridge, MA 02139, USA

**Keywords:** network biology, ChIP-seq, breast cancer, transcription factors, EMT, functional gene network, mesoderm, *Drosophila melanogaster*, gene regulation, epithelial remodelling

## Abstract

Cell identity is governed by gene expression, regulated by transcription factor (TF) binding at cis-regulatory modules. Decoding the relationship between TF binding patterns and gene regulation is nontrivial, remaining a fundamental limitation in understanding cell decision-making. We developed the NetNC software to predict functionally active regulation of TF targets; demonstrated on nine datasets for the TFs Snail, Twist, and modENCODE Highly Occupied Target (HOT) regions. Snail and Twist are canonical drivers of epithelial to mesenchymal transition (EMT), a cell programme important in development, tumour progression and fibrosis. Predicted “neutral” (non-functional) TF binding always accounted for the majority (50% to 95%) of candidate target genes from statistically significant peaks and HOT regions had higher functional binding than most of the Snail and Twist datasets examined. Our results illuminated conserved gene networks that control epithelial plasticity in development and disease. We identified new gene functions and network modules including crosstalk with notch signalling and regulation of chromatin organisation, evidencing networks that reshape Waddington’s epigenetic landscape during epithelial remodelling. Expression of orthologous functional TF targets discriminated breast cancer molecular subtypes and predicted novel tumour biology, with implications for precision medicine. Predicted invasion role*s* were validated using a tractable cell model, supporting our approach.

## 1. Introduction

Transcriptional regulatory factors (TFs) govern gene expression, which is a crucial determinant of phenotype. Mapping transcriptional regulatory networks is an attractive approach to understand the molecular mechanisms underpinning both normal biology and disease [1,2,3]. TF action is controlled in multiple ways; including protein–protein interactions, DNA sequence affinity, 3D chromatin conformation, post-translational modifications and the processes required for TF delivery to the nucleus [3,4,5]. A complex interplay of mechanisms influences TF specificity across different biological contexts and genome-scale assignment of TFs to individual genes is challenging [1,5,6].

TF binding sites may be determined using chromatin immunoprecipitation followed by sequencing (ChIP-seq) or microarray (ChIP-chip). These and related methods (e.g., ChIP-exo, DamID) have revealed a substantial proportion of statistically significant “neutral” TF binding, that has apparently no effect on transcription from assigned target genes [1,7,8,9]. Genomic regions that bind large numbers of TFs, termed Highly Occupied Target (HOT) regions [10], are enriched for disease SNPs and can function as developmental enhancers [11,12]. However, a considerable proportion of individual TF binding events at HOT regions may have little effect on gene expression and association with chromatin accessibility suggests non-canonical regulatory function such as sequestration of TFs or in 3D genome organization [13,14], as well as possible technical artefacts [15]. Apparent neutral binding events may also have subtle functions; for example, in combinatorial context-specific regulation or in buffering noise [2,16]. While recent integrative work enhances context-independent TF target prediction [17], identification of bona fide functional TF target genes remains a major obstacle in understanding the regulatory networks that control cell behaviour [2,5,9,18,19].

Genes regulated by an individual TF typically have overlapping expression patterns and coherent biological function [20,21,22]. Indeed, gene regulatory networks are organised in a hierarchical, modular structure and TFs frequently act upon multiple nodes of a given module [23,24]. Therefore, we hypothesised that the network properties of functional TF targets are different to those of neutrally bound sites. Network analysis can reveal biologically meaningful gene modules, including cross-talk between canonical pathways [25,26,27] and so may enable elimination of neutrally bound candidate TF targets derived from statistically significant ChIP-seq or ChIP-chip peaks. Network approaches afford significant advantages for handling biological complexity, enable genome-scale analysis of gene function [28,29], and are not restricted to predefined gene groupings used by standard functional annotation tools (e.g., GSEA, DAVID) [25,30,31]. Clustering is frequently applied to define biological modules [32,33]. However, pre-defined modules may miss condition-specific features; for example, gene products may be absent in the biological condition(s) analysed but included in pre-defined network modules. Hence, clusters derived from a whole-genome network may not accurately capture biological interactions that occur in a particular context. Context-specific interactions are common, for example the varied repertoire of biophysical interactions in different cell types or between cell states, such as in the stages of the cell cycle [34]. We developed an algorithm (NetNC) with capability for context-specific functional TF target discovery and applied this to study the epithelial to mesenchymal transition (EMT) TFs Snail and Twist. EMT is a multi-staged morphogenetic programme fundamental for normal embryonic development that contributes to tumour progression and fibrosis [35,36,37,38]. We predicted Snail and Twist functional targets, integrating these predictions with results from genetic screens and breast cancer transcriptomes; in order to study epithelial remodelling in development and disease.

## 2. Results and Discussion

### 2.1. A Comprehensive Drosophila melanogaster Functional Gene Network (DroFN)

Our approach requires a genome-scale map of gene function; for this purpose we developed the DroFN network (11,432 genes; 787,825 interactions). DroFN models *Drosophila melanogaster* signalling and metabolism, integrating the Gene Ontology (GO) [39] and STRING [40] databases, calibrated against KEGG pathways using Bayesian Logistic Regression [41]. DroFN performed well on blind test data (TEST-NET) compared with other *D. melanogaster* gene networks (DroID [42], GeneMania [43]) (Appendix A, Appendix A). The positive class in TEST-NET was formed from gene pairs within the same KEGG pathway and the negative class was derived from gene pairs with no evidence for a pathway interaction. The overlap between DroFN and the *Drosophila* proteome interaction map (DPiM [44]) was highly significant (Fisher Exact Test *p* < 10^−308^). DroFN and DPiM had 999 genes in common and 37.8% (2175/5747) of DroFN interactions for these genes were also found in DPiM. The DroFN false positive rate (0.043) was close to the prior expectation for functional gene interactions (0.044); thus, a proportion of these apparent false positives might represent bona fide interactions missing from the gold standard KEGG pathways. Overall, DroFN provides a comprehensive, high-quality representation of pathway comembership in *D. melanogaster*.

### 2.2. Prediction of Functional Transcription Factor Targets

We present NetNC, an algorithm for genome-scale prediction of functional TF target genes (Figure 1). NetNC built upon observations that TFs coordinately regulate multiple functionally related targets [20,21,22] and was calibrated for discovery of biologically coherent genes in noisy data, according to the structure of a gene network. This approach required optimisation for elimination of noise in biological data, rather than for community detection. Statistical evaluation of network coherence for an input gene list, including false discovery rate (FDR) estimation, is applied within NetNC as basis for numerical thresholding. Therefore, NetNC can analyse single-subject datasets, which is an important emerging area for precision medicine [45]; for example, to derive networks from genes with high or low relative expression according to ranked expression values from a single sample. Application of statistical and graph theoretic methods in NetNC for quantitative evaluation of relationships between genes offers an alternative to the classical emphasis on individual genes in studying the relationship between genotype and phenotype.

NetNC has two different analysis settings, NetNC-FTI (“Functional Target Identification”) and NetNC-FBT (“Functional Binding Target”), shown in Figure 1. Biological similarity between gene pairs is represented in NetNC using shared network neighbours, formalised by the Hypergeometric Mutual Clustering coefficient [46]; further analysis steps then enable prediction of functional TF targets (Figure 1). The current study reports results from NetNC with DroFN as a reference network, however NetNC may be used to analyse any network and node list. We chose the DroFN network because of its favourable performance in comparisons against DroID and GeneMania (Appendix A, Appendix A). In order to assess NetNC performance and to calibrate algorithm parameters, we developed gold standard data using KEGG pathways and “Synthetic Neutral Target Genes” (SNTGs). Clustering coefficient (CC) values predicted by models trained on the synthetic benchmark data (methods Section 3.7) matched the CC values calculated directly on the nine TF_ALL datasets, which are described in methods Section 3.11. For 8/9 datasets the difference between predicted and actual CC was <0.1 (median CC difference = 0.051, 95% CI 0.007–0.136). This similarity in CC values for the synthetic and biologically-derived candidate TF target genes supports the application of our benchmark in the context of network-based functional TF target prediction. NetNC was robust to variation in input dataset size and %SNTGs, outperforming the clustering algorithms HC-PIN [33] and MCL [32] (Figure 2, Appendix A). In general, NetNC was more stringent, with lower false positive rate (FPR) and higher Matthews Correlation Coefficient (MCC). MCC provided a measure of overall performance in correctly separating functional targets from neutral binding (SNTGs). FPR indicated the proportion of predicted functional targets that were SNTGs and therefore classified incorrectly. At the highest %SNTG, NetNC-FTI overall performance (MCC) was around 50% to 67% better than HC-PIN, and NetNC-FBT typically had lowest FPR. The task of separating SNTGs from all of the genes that form pathways is subtly different to cluster identification; thus neither HC-PIN nor MCL were developed for the precise application evaluated here. A number of network-based analysis method such as HotNet2 [47] and PrixFixe [48] were developed for very different application areas, for example requiring mutational frequency data, and so were not suitable for application to functional TF target discovery. While NetNC-FTI performed best overall, NetNC-FBT is parameter-free, did not require training and therefore may be more robust for analysis of diverse input datasets and reference networks other than DroFN. NetNC’s performance advantages were most prominent on data with ≥50% SNTGs (Figure 2) and predicted neutral binding for TF_ALL was ≥61% (Figure 3, Table 1) or ≥50% using an alternative calibration method (Appendix A). 

Given the performance advantage at ≥50% STNGs, NetNC appears the method of choice for analysis of genome-scale TF occupancy data. However, NetNC may also be applied to analyse various data-types, including in: identification of differentially expressed pathways and macromolecular complexes from functional genomics data; illuminating common biology among CRISPR screen hits in order to inform prioritisation of candidates for follow-up work [49]; and discovery of functional coherence in chromosome conformation capture data (4-C, 5-C), for example in enhancer regulatory relationships [50,51]. We also compared NetNC against the NEST algorithm [52] and against node degree (Appendix A). The NEST output required filtering and did not provide a threshold for separation of predicted neutral binding from functional targets, therefore we compared Area Under the Receiver Operator Characteristic curve (AUC). There was a trend towards NetNC having better performance than the filtered NEST results; predictions that took either node degree or the unfiltered NEST output had substantially lower AUC values than NetNC. NetNC results were robust to subsampled input gene lists (Appendix A). Gold standard datasets and DroFN are available from the BioStudies database: www.ebi.ac.uk/biostudies/studies/S-BSST460. NetNC is available from https://github.com/overton-group/NetNC.

### 2.3. Analysis of EMT Transcription Factors and Highly Occupied Target (HOT) Regions

We predicted functional target genes for the Snail and Twist TFs in developmental stages around *D. melanogaster* gastrulation. Chromatin ImmunoPrecipitation (ChIP) microarray (ChIP-chip) or sequencing (ChIP-seq) data from four different laboratories were analysed for overlapping time periods in early embryogenesis [8,22,53,54]. Two of these datasets were from studies examining the regulatory networks that control dorsoventral patterning, including mesoderm development [53,54]. Further Snail and Twist binding data was available from a comparatively large investigation of patterning in early fly development by the Berkley Drosophila Transcriptional Network Project, which analysed 21 TFs including dorsoventral factors [22]. We also obtained data from more recent work that compared ChIP-seq to other technologies for genome-scale analysis of gene regulation, including identification of new Twist-occupied regions and a consensus motif analysis [8]. An additional dataset examined TF binding hotspots termed “Highly Occupied Target” (HOT) regions that were annotated by the modENCODE project [10]. Overall, nine main datasets were studied (TF_ALL, Table 1; please see Methods Section 3.11 for further details), the proportion of predicted functional TF binding for these datasets ranged from 5% to 39% (Figure 3A, Table 1). At the time of writing, the above datasets remain leading sources of information for Snail and Twist binding in early mesoderm development. NetNC does not perform peak calling—but is intended for downstream analysis of candidate target genes from statistically significant peaks, in order to enable separation of predicted functional targets from neutral binding. In addition to the predictions from NetNC-FTI, we developed a complementary approach to estimate the total functional binding in each TF_ALL dataset; this method was based on local FDR (NetNC-lcFDR; Methods, Section 3.4).

Reassuringly, candidate TF targets from the most stringent peak calling approach (twi_1–3h_hiConf [8]) had comparatively high predicted functional binding (PFB). Despite having high PFB, twi_1–3h_hiConf had the smallest proportion of genes passing pFDR < 0.05 (Figure 3D). High PFB was also found for targets bound during two consecutive developmental time periods (twi_2–6h_intersect [53]), as well as for HOT regions despite their lack of known TF motifs [11,55,56] (Figure 3A, Table 1). Indeed, twi_2–6h_intersect had significantly higher PFB (binomial *p <* 4.0 × 10^−15^) than datasets from the same study that represented a single time period (twi_2–4h_intersect, twi_4–6h_intersect) [53] (Figure 3). Therefore, PFB was enriched for regions occupied at >1 time period or by multiple TFs and results supported the emerging picture of widespread combinatorial control involving TF–TF interactions, cooperativity and TF redundancy [2,5,57,58,59]. PFB was similar for sites derived from either the union or intersection of two Twist antibodies, although the NetNC-FTI method found a higher number of functional targets for the intersection of antibodies (30.5% (116/334) vs. 23% (424/1848)). Hits identified by multiple antibodies may be technically more robust due to reduced off-target binding [53]. However, taking the union of candidate binding sites could eliminate false negatives arising from epitope steric occlusion due to protein interactions. The similarity in PFB for either the intersection or the union of Twist antibodies suggests that, despite expected higher technical specificity, the intersection of candidate targets may not enrich for functional binding sites at the 1% peak-calling FDR threshold applied in [22,53]. Fewer false negatives implies recovery of numerically more functional TF targets, likely producing denser clusters in DroFN, which could facilitate functional target detection by NetNC. Indeed, datasets representing the union of two antibodies ranked highly in terms of both the total number and proportion of genes recovered at lcFDR < 0.05 or pFDR < 0.05 (Figure 3). Even datasets with low PFB had candidate target genes that passed stringent NetNC FDR thresholds; for example, sna_2–3h_union, twi_2–3h_union respectively had the highest and second-highest proportion of candidate targets at lcFDR < 0.05 (Figure 3B). We found no evidence for benefit in using RNA polymerase binding data to guide allocation of peaks to candidate target genes (datasets sna_2–3h_union, twi_2–3h_union). The twi_2–4h_Toll^10b^, sna_2–4h_Toll^10b^ datasets had a relatively low peak threshold (two-fold enrichment), which may have contributed to the low PFB for sna_2–4h_Toll^10b^. We note that our analysis might systematically overestimate neutral binding because some functional targets could be missed; for example, due to errors in assigning enhancer binding to target genes and in *bona fide* regulation of genes that have few DroFN edges with other candidate targets. Predicted neutral targets for twi_2-4_intersect, twi_2-6_intersect and twi_4-6_intersect were overwhelmingly unchanged in *Twist* loss-of-function gene expression data from the same study [53] (respectively 96–97%, 93–94%, 90–91%, and 93–95% were unchanged at the 4–5 h, 5–6 h, 6–7 h, and 7–8 h time points; *q* < 0.05, 1.5 FC). We also note that NetNC-lcFDR and NetNC-FTI neutral binding estimates showed good agreement (Table 1, Appendix A).

ChIP peak intensity putatively correlates with functional binding, although some weak binding sites are functional [9,60]. The NetNC Node Functional Coherence Score (NFCS) and ChIP peak enrichment scores were significantly, although weakly, correlated in 6/8 datasets (*q* < 0.05, HOT regions not analysed; median rho = 0.11). Datasets with no significant correlation (twi_1–3h_hiConf, twi_2–6h_intersect) derived from protocols that enriched for functional targets and had highest PFB (Figure 3A). Indeed, the median peak score for twi_2–6h_intersect was significantly higher than datasets taken from a single time period in same study (twi_2–4h_intersect, *q* < 5.0 × 10^−56^; twi_4–6h_intersect, *q <* 4.8 × 10^−58^). The number of orthologues for each dataset correlated strongly with the number of predicted functional targets (r = 0.973). However, sna_2–3h_union and twi_2–3h_union functional targets had significantly higher proportion of orthologues (>80%) than the next highest dataset (twi_2–4h_intersect, 67%; respective binomial test *q* < 3.9 × 10^−4^, *q* < 1.2 × 10^−11^), which might be explained by the use of RNA polymerase binding data in assigning candidate target genes. NetNC-FTI predictions were enriched for human orthologues relative to respective candidate target genes in DroFN (Appendix A), for example predicted twi_2–3h_union functional targets had 82% (347/424) human orthology vs. 61% (1135/1848) for the pool of candidate targets (binomial *q* < 8.7 × 10^−19^). Annotation bias might contribute something to this significant enrichment, because conserved genes are more deeply studied and may associated with a higher degree in the DroFN network; high degree is also expected for conserved genes because of their functional importance [61]. We note that enrichment for evolutionary conservation in Snail and Twist functional targets aligned with their regulation of fundamental developmental processes [35,36].

### 2.4. Genome-Scale Functional Transcription Factor Target Networks

NetNC results offered a global representation of tissue-specific regulation by Snail and Twist in early *D. melanogaster* embryogenesis (Appendix A, Data File S1). Results revealed 11 biological groupings common to ≥4/9 TF_ALL datasets (Appendix A). We found Snail and Twist regulation of multiple core cell processes that govern the global composition of the transcriptome and proteome, including: transcription, chromatin organisation, RNA splicing, translation and protein turnover. These predicted regulatory events may contribute to either repression or activation of individual genes in the (presumptive) mesoderm. A “Developmental Regulation Cluster” (DRC) was identified in every TF_ALL dataset and contained members of multiple key conserved morphogenetic pathways, including notch and wnt. We examined predicted functional targets that were found in the DRC and chromatin organisation clusters for multiple TF_ALL datasets (Figure 4A); *Wingless* had the highest degree and strongest edges in the combined network; *Notch* and *forkhead* had joint second highest frequency, represented in 8/9 NetNC-FTI results for TF_ALL. Many of the DRC genes were previously reported to be important for mesoderm development [53,62] and their interactions suggest functional relationships. For instance, genes that interact with *T48* might contribute to *fog*-independent ventral furrow formation [63]. The edge between *Notch* and *wingless* was identified most frequently in the combined DRC network. *Notch* signalling modifiers identified in at least two public datasets [64] overlapped significantly with the overall NetNC results for each TF_ALL dataset (*q* <0.05), including members of the DRC, chromatin organisation and mediator complex clusters (Figure 4, Appendix A). Activation of *Notch* can result in diverse, context-specific transcriptional outputs and the mechanisms regulating this pleiotropy are not well understood [64,65,66,67]. Our results provided functional context for many *Notch* modifiers and proposed signalling crosstalk mechanisms in cell fate decisions driven by Snail and Twist, where regulation of modifiers may control the consequences of *Notch* activation. Crosstalk between *Notch* and *twist* or *snail* was previously shown in multiple systems, for example in adult myogenic progenitors [68] and hypoxia-induced EMT [69]. Consistent with previous studies [64,65], our results predicted targeting of *Notch* transcriptional regulators, trafficking proteins, post-translational modifiers, receptor recycling, and regulation of pathways that may attenuate or modify *Notch* signalling. Clusters where multiple modifiers were identified may represent cell meso-scale units important for *Notch* in the context of mesoderm development and EMT (Appendix A). For example, the mediator complex and transcription initiation subcluster for twi_2–3h_union had 13 nodes, of which five were *Notch* modifiers including orthologues of *MED7*, *MED8*, and *MED31* (Data File S1). These results highlight key control points regulated by Snail, Twist in fly mesoderm specification; including *wingless*, *forkhead*, and *Notch*. Thirteen DRC genes were present in ≥7 TF_ALL datasets (DRC-13, Appendix A), and had established functions in the development of mesodermal derivatives such as muscle, the nervous system and heart [66,68,70,71,72,73,74]. Supporting NetNC predictions, in situ hybridisation for DRC-13 genes indicated expression in (presumptive) mesoderm at: Stages 4–6 (*wg*, *en*, *twi*, *N*, *htl*, *how*), stages 7–8 (*rib*, *pyd*, *mbc*, *abd-A*) and stages 9–10 (*pnt*) [75,76,77]. The remaining two DRC-13 genes had no evidence for mesodermal expression (*fkh*) or no data available (*jar*). However, *fkh* was essential for caudal visceral mesoderm development [78] and *jar* was expressed in midgut mesoderm [79].

Networks produced by NetNC-FTI for each of the nine TF_ALL datasets frequently included chromatin organisation clusters (Appendix A and Appendix A); recurrently identified nodes from these clusters corresponded to trithorax-group (TrxG) and polycomb-group (PcG) genes which exert dynamic, opposing gene-regulatory activity [80] (Figure 4B). The PcG cluster had Polycomb Repressive Complex 1 (PRC1) genes *ph-d*, *ph-p*, and *Psc* [81], the gene silencing factor S*u(var)3–9* [82], as well as *corto* [83] and *lolal* [84] (Figure 4B, Appendix A). These results predict that *corto* and *lolal* function in concert with core PRC1 members in *D. melanogaster* mesoderm development under the control of Twist and Snail. Indeed, interaction with accessory proteins enables context-specific PRC1 function, and would merit further study in *Drosophila* [80]. The TrxG cluster contained *tara* [85], *trx* [86], *tna* [87], *mor* [88], s*u(var)205* [89], *Bgb* [90], and *JIL-1* [91]. *Tara* and *tna* were predicted functional targets in 6/9 datasets, interact with each other and with the *brahma* chromatin remodelling complex [87]. A third cluster was formed from chromatin assembly factors, including the poorly characterised gene CG3708 that is orthologous to the nucleosome assembly protein NAP1L1. PcG genes are crucial oncofetal regulators and the focus of significant cancer drug development efforts [92,93]. These results align with reports that gene silencing in EMT involved PcG [93,94] and with *Snai1* recruitment of Polycomb Repressive Complex 2 members [94], supporting a model where EMT TFs control the expression of their own coregulators.

Snail regulation of neural genes (Appendix A, Appendix A) was consistent with its repression of ectodermal (neural) genes in the prospective mesoderm [62,95,96]. Indeed, clusters relating to brain development were found in six TF_ALL datasets. Additionally, Snail is important for neurogenesis in fly and mammals [97,98]. Therefore, binding to neural functional modules might reflect potentiation for rapid activation in combination with other transcription factors within neural developmental trajectories [53,99]. NetNC results predicted novel Twist functions, for example in activation or repression of mushroom body neuroblast proliferation factors Rx, sle, and tara. The mushroom body is a prominent structure in the fly brain, important for olfactory learning and memory [100], identified in analysis of six TF_ALL datasets (Appendix A). Twist is typically a transcriptional activator [96] although may contribute to Snail’s repressive activity [101]. Indeed, TWIST1 repressed Cadherin-1 in breast cancers [102].

### 2.5. Breast Cancer Subtypes are Recovered by Unsupervised Clustering with Orthologous Snail and Twist Functional Targets

We analysed the conserved molecular networks that orchestrate epithelial remodelling in development and cancers by combining NetNC results for TF_ALL with the results of *Notch* screens and breast cancer transcriptomes; data integration was based on identifier matching and orthology mapping (methods). Predicted Snail and Twist targets included known cancer genes and also suggested novel drivers (Appendix A, Appendix A, Appendix A and Appendix A). The fly genome is relatively tractable for network studies, while data availability (e.g., ChIP-chip, ChIP-seq, genetic screens) is enhanced by both considerable community resources and the relative ease of experimental manipulation [103]. Many developmentally patterned fly genes are orthologues of established cancer drivers. Breast cancer intrinsic molecular subtypes with distinct clinical trajectories were extensively validated and complement clinico-pathological parameters [104,105]. These subtypes are known as luminal-A, luminal-B, HER2-overexpressing, normal-like, and basal-like. While more recent studies have classified further subtypes, for example identifying ten groups [106], the five subtypes employed in our analysis had been widely used, extensively validated, exhibited clear differences in prognosis, overlapped with subgroups defined using standard clinical markers (*ESR1*, *HER2*), and aligned with distinct treatment pathways [104,105]. The NetNC-FTI networks for all nine TF_ALL datasets overlapped with known cancer pathways, including significant enrichment for *Notch* modifiers (*q* < 0.05). We hypothesised that orthologous genes from NetNC clusters for Snail and Twist would stratify breast cancers by intrinsic molecular subtype. Indeed, aberrant activation of *Notch* orthologues in breast cancers had been demonstrated, and linked with EMT-like signalling, particularly for basal-like and claudin-low subtypes [107,108,109]. One might expect the predicted Snail and Twist functional targets to be prognostic in multiple different cancers due to the representation of established cancer cell processes, for example DNA replication and repair as well as developmental regulation (Appendix A, Appendix A).

We hypothesised that orthologous genes from Snail and Twist functional targets would stratify breast cancers into clinically meaningful groups. Sixty-eight DroFN genes were predicted functional targets in four or more of the nine TF_ALL datasets and also had human orthology. Fifty-seven of these sixty-eight genes (ORTHO-57) were represented in integrated gene expression microarray data for 2999 breast tumours (BrC_2999) [110]. Taking a threshold of at least four datasets reduced the gene list size used for clustering in order to help prevent stratification effects arising from autocorrelation [111]. Unsupervised clustering using ORTHO-57 and also NetNC results for individual Twist, Snail datasets stratified BrC_2999 by intrinsic molecular subtype (Figure 5, Appendix A). Predicted functional targets had significantly higher centroid values than equivalent resampled orthologues from DroFN, demonstrating that the observed stratification of breast cancer subtypes was significantly greater than expected by chance (Appendix A). Previous work suggested roles in tumour biology for many ORTHO-57 genes; however, few were linked to an intrinsic breast cancer subtype. Searching PubMed with the gene name(s) and “breast cancer” found little evidence of function in breast cancer invasion for 13/57 genes (ADSS, CREG1, ATP5A1, SRSF2, SNRPD1, RNPS1, TEC, HIVEP3, SERTAD2, NACC2, GULP1, IRX4, and TRIB2). Heatmap features were annotated as dashed black boxes according to the dendrogram structure and gene expression intensity (Figure 5). The datasets sna_2–4h_Toll^10b^, twi_2–4h_Toll^10b^ represented embryos formed entirely from mesodermal lineages [54] and, together, had significantly greater proportion of basal-like breast cancer genes than the combined sna_2–3h_union, twi_2–3h_union datasets (*p* < 8.0 × 10^−4^); consistent with EMT characteristics of basal-like breast cancers [112]. As expected, Basal-like tumours were characterised by *EN1* and *NOTCH1* [107,108,113]. Notch signalling modulation is a promising area for cancer therapy [65] and *Notch* modifier orthologues from our analysis could potentially inform development of companion diagnostics or combination therapies targeting the notch pathway in basal-like breast cancers. Elevated *ETV6* expression was also a feature of basal-like cancers, where copy number amplifications and recurrent gene fusions were previously reported [114,115]. The Luminal A subtype (feature_LumA) had similarities with luminal B (feature_LumB_2_, ERBB3, MYO6) and normal-like (DOCK1, ERBB3, MYO6) tumours. High *BMPR1B* expression was the major defining feature for luminal A tumours, aligning with oncogenic BMP signalling in luminal epithelia [116]. However, BMP2 expression was highest in basal-like cancers, where it may drive an EMT programme [117]. Several genes were highly expressed in both Luminal B and *ESR1* negative subtypes (feature_LumB_1_, feature_ERneg) including *ECT2*, *SNRPD1*, *SRSF2*, *CBX3*; these genes might contribute to worse survival outcomes for luminal B relative to luminal A cancers [104,118]. Indeed, analysis with the GEPIA2 resource [119] revealed that these four genes stratified breast cancer patients by overall survival in an independent cohort from The Cancer Genome Atlas [120], where high expression conferred worse prognosis (Appendix A). Feature_LoExp represented genes with low detection rates across a mixture of subtypes, largely from a single study [121]. Key EMT genes (*SNAI2*, *TWIST1*, and v*QKI*) were assigned to the NL centroid and had highest relative expression in normal-like tumours (feature_NL, Figure 5). Feature_NL also included homeobox transcription factors (*HOXA9, MEIS2*) and a secreted cell migration guidance gene (*SLIT2*). Genes with high expression in both normal-like and basal-like cancers included *QKI*, which regulates circRNA formation in EMT [122], and the *FZD1* wnt/β-catenin receptor. Moreover, genes in feature_Bas and feature_NL clustered together, identifying similarities between normal-like and basal-like subtypes. EMT may confer stem-like cell properties [123,124,125] and our results were consistent with dedifferentiation or arrested differentiation due to activation of an EMT-like programme in NL cancers. Previous work found stem cell markers in NL cancers [118,126], indeed *SNAI2* was important in both mammary and breast cancer stem cells [127,128]. However, high stromal content in NL tumours [129] might also contribute to an EMT-like gene expression signature. In summary, the predicted functional TF targets from fly mesoderm development captured clinically relevant molecular features of breast cancers and proposed candidate subtype-specific drivers of tumour progression.

### 2.6. Integrating NetNC Functional Target Networks and Breast Cancer Transcriptome Profiling

Orthologous basal-like and normal-like genes were annotated onto NetNC-FTI networks, offering a new perspective on the molecular circuits controlling these different subtypes (Appendix A). Interestingly, key EMT genes were assigned to the normal-like subtype, which was also associated with splicing factors, the ribosome, the proteasome and proteasome regulatory subunits. The sna_2–4h_Toll^10b^ “RNA degradation and transcriptional regulation” cluster was exclusively annotated to the basal-like subtype, including *HECA*, which was upregulated in basal-like relative to normal-like tumours (*p* < 3.3 × 10^−23^). NetNC also identified the fly orthologue of *HECA*, *hdc*, in both twi_2–4h_intersect and twi_4–6h_intersect, bound at non-contiguous sites. *Hdc* was a multifunctional *Notch* signalling modifier, including in cell survival [130] and tracheal branching morphogenesis, activated by *escargot* [131]. Taken together, these data support participation of *HECA* in an EMT-like gene expression programme in basal-like breast cancers. The *SLC9A6* Na^+^/H^+^ antiporter was found in NetNC-FTI ion transport clusters for sna_2–4h_Toll^10b^ and twi_2–4h_Toll^10b^. Alterations in pH by Na^+^/H^+^ exchangers, particularly *SLC9A1*, drive basal-like breast cancer progression and chemoresistance [132]. *SLC9A6* was upregulated in basal-like relative to normal-like tumours (*p* < 8.4 × 10^−71^) and might cause pH dysregulation as part of an EMT-like programme in basal-like cancers.

Chromatin organisation clusters frequently associated with basal-like annotations. For example, the twi_2–3h_union “chromatin organisation and transcriptional regulation” cluster had six basal-like genes, including three Notch modifiers (*ash1*, *tara*, *Bap111*). These were orthologous to the *ASH1L* histone methyltransferase that had copy number amplifications in basal-like tumours [133]; the *SERTAD2* bromodomain interacting oncogene and E2F activator [134]; and *SMARCE1*, a core subunit of the SWI/SNF chromatin remodelling complex that regulated *ESR1*, interacted with *HIF1A* signalling and potentiated breast cancer metastasis [135,136,137]. Notch can promote EMT-like characteristics and mediated hypoxia-induced invasion in multiple cell lines [69]. Consistent with these studies, our work supported conserved function for *SMARCE1* in EMT-like signalling, both in mesoderm development and basal-like breast cancers, possibly downstream of *NOTCH1* and through regulation of SWI/SNF targeting. Indeed, SWI/SNF controlled chromatin switching in oral cancer EMT [138]. *Taranis*, orthologous to *SERTAD2*, also functioned to stabilise the expression of *engrailed* in regenerating tissue [85]. The *engrailed* orthologue *EN1* was the clearest single basal-like biomarker in the data examined (Figure 5) and acted as a survival factor [113]. *SERTAD2* and *EN1* expression values correlated in the basal-like tumours (*n* = 573, *p* < 2.0 × 10^−9^, rho = 0.25) but not across the entire cohort (*n* = 2999, *p* = 0.44, rho = −0.03). Our results suggest that *SERTAD2* could cooperate with *EN1* in a subset of basal-like cancers, where coordinated expression of these two genes may form part of a gene expression programme controlled by EMT TFs. Regulation of *EN1*, *SERTAD2* within an EMT programme could harmonise previous results demonstrating key roles for both neural-specific and EMT TFs in basal-like breast cancers [112,113]. Therefore, our results highlight chromatin organisation factors downstream of Snail and Twist with orthologues that may control *Notch* output and breast cancer progression through a chromatin remodelling mechanism. Indeed, NetNC results predicted components of feedback loops where EMT TFs regulate chromatin organisation genes that, in turn, may both reinforce and coordinate downstream stages of gene expression programmes for mesoderm development and cancer progression. Stages of the EMT programme were described elsewhere, reviewed in [35]; our results mapped networks that may control the remodelling of Waddington’s landscape—identifying crosstalk between Snail, Twist, epigenetic modifiers and regulation of key developmental pathways [139]. Dynamic interplay between successive cohorts of TFs and chromatin organisation factors is an attractive potential mechanism to determine progress through and the ordering of steps in (partial) EMTs, consistent with “metastable” intermediate stages.

### 2.7. Novel Twist and Snail Functional Targets Influence Invasion in a Breast Cancer Model of EMT

NetNC results predicted new gene functions in EMT and cell invasion. We investigated the functional and instructive role of four genes in an established invasion model [140]; *SNX29* (also known as *RUNDC2A*), *ATG3*, *IRX4,* and *UNK*. These genes were selected for experimental follow-up from the large pool of NetNC results for TF_ALL according to novelty in the context of cell invasion and EMT. The orthologues of *IRX4* and *SNX29* were predicted to be regulated by both Snail and Twist, the orthologue of *ATG3* was predicted to be a functional target of Twist; and the orthologue of *UNK* was predicted to be regulated by Snail. MCF7 cells were weakly invasive [141], thus the *SNAI1*-inducible MCF7 cell line was well suited to study alteration in expression of the selected genes in terms of their influence on invasion in conjunction with *SNAI1* induction, knockdown or independently (Figure 6).

Over-expression of *IRX4* significantly increased invasion relative to controls in all conditions examined and *IRX4* had high relative expression in a subset of basal-like breast cancers (Figure 5 and Figure 6). *IRX4* was a homeobox transcription factor involved in cardiogenesis, marking a ventricular-specific progenitor cell [142] and was also associated with prostate cancer risk [143]. *SNX29* was poorly characterised, belonged to the sorting nexin protein family that function in endosomal sorting and signalling [144], and ectopic expression significantly reduced invasion in a *SNAI1*-dependent manner (Figure 6). Since we obtained these results, *SNX29* downregulation was associated with metastasis and chemoresistance in ovarian carcinoma [145], consistent with *SNX29* inhibition of invasion driven by Snail. *ATG3* was an E2-like enzyme required for autophagy and mitochondrial homeostasis [146], *ATG3* overexpression significantly increased MCF7 invasion. Knockdown of *ATG3* reduced invasion in hepatocellular carcinoma [147]. *UNK* was a RING finger protein homologous to *unkempt* which bound mRNA, functioned in ubiquitination and was upregulated in gastrulation [148]. Others reported that UNK mRNA binding controlled neuronal morphology and induced spindle-like cell shape in fibroblasts [149,150]. UNK significantly increased MCF7 invasion both independently of and additively with Snail; supporting a potential role in breast cancer progression. Indeed, UNK was overexpressed in cancers relative to controls in ArrayExpress [151]. These in vitro confirmatory results both supported the novel analysis approach and evidenced new function for the genes examined.

## 3. Methods

### 3.1. A Comprehensive D. Melanogaster Functional Gene Network (DroFN)

A high-confidence, comprehensive *Drosophila melanogaster* functional network (DroFN) was developed using a previously described supervised learning approach in order to model global gene function [152]. Genes were network nodes and their interactions represented associations within biological pathways. Gene interactions were quantified by functional interaction probabilities, reflecting pathway co-membership, estimated by logistic regression of Bayesian probabilities from STRING v8.0 scores [40] and Gene Ontology (GO) coannotations [39]; KEGG [41] pathways were taken as gold standard.

Gene pair co-annotations were derived from the GO database of 25th March 2010. The GO Biological Process (BP) and Cellular Component (CC) branches were read as a directed graph and genes added as leaf terms. The deepest term in the GO tree was selected for each gene pair, and BP was given precedence over CC. Training data were taken from KEGG v47, comprising 110 pathways (TRAIN-NET). Positive gold standard gene pairs were derived from genes found within the same pathway, the remaining gene pairs had no evidence for a pathway comembership interaction and were therefore assigned to the negative class. Bayesian probabilities for STRING and GO coannotation frequencies were derived from TRAIN-NET [152]. Selection of non-interacting “negative” pairs from TRAIN-NET using the *perl* rand() function was used to generate training data with equal numbers of positive and negative pairs (TRAIN-BAL). This approach to selection of negative pairs by random sampling helps to avoid bias [153]. TRAIN-BAL which was input for logistic regression, to derive a model of gene pair functional interaction probability (Equation (1)):(1)$$p\left(I|GO,STRING\right)=\frac{1}{1+\left(e^{-6.75+1.03pGO+1.12pSTRING}\right)}$$
where: pGO is the Bayesian probability derived from Gene Ontology coannotation frequency, pSTRING is the Bayesian probability derived from the STRING score frequency.

The above model was applied to TRAIN-NET and the resulting score distribution thresholded by seeking a value that maximised the F-measure [154] and true positive rate (TPR), while also minimising the false positive rate (FPR). The selected threshold value (*p* ≥ 0.779) was applied to functional interaction probabilities for all possible gene pairs to generate the high-confidence network, DroFN.

For evaluation of the DroFN network, time separated test data (TEST-TS) were taken from KEGG v62 on 13/6/12, consisting of 14 pathways that were not in TRAIN-NET. The pathways in the time separated test dataset were not present in KEGG at the time when the training data was downloaded, supporting stringent independent evaluation; indeed, this principle is employed in community critical assessment work [155]. Gene pairs were eliminated from TEST-TS if either gene was found in TRAIN-NET, removing 76 positive and 1294 negative gene pairs to generate the blind test dataset TEST-NET (3481 pairs). Therefore, all gene pairs in TEST-NET corresponded to the most stringent evaluation class, termed “C3” by Park and Marcotte [156]. The most up-to-date versions of GeneMania (v2017-03-14) [43] and DroID (v2018_08) [42] downloaded April 2020 were assessed against TEST-NET (Appendix A, Appendix A).

Enrichment of DroFN edges in DPiM [44] was estimated as follows: A total of 999 genes were found in both DroFN and the DroPIM network thresholded at FDR 0.05 (DroPIM_FDR). These 999 genes had 5747 edges in DroPIM_FDR and 25,797 edges in DroFN, of which 2175 were common to both networks. A 2 × 2 contingency table was constructed conditioning on the presence of edges for these 999 genes in the DroFN and DroPIM_FDR networks. The contingency table cell corresponding to edges not found in DroFN or DroPIM was populated by the number of possible edges for the 999 genes ((*n*^2^−*n*)/2), subtracting the values from the other cells. Therefore, the contingency table cell values were: 2175, 3572, 23,622, 469,132. The enrichment *p*-value was calculated by Fisher’s Exact Test.

### 3.2. Network Neighbourhood Clustering (NetNC) Algorithm

NetNC identifies functionally coherent nodes in a subgraph *S* of functional gene network *G* (an undirected graph), induced by some set of nodes of interest *D*; for example, candidate transcription factor target genes assigned from analysis of ChIP-seq data. Intuitively, we consider the proportion of common neighbours for nodes in *S* to define coherence; for example, nodes that share neighbours have greater coherence than nodes that do not share neighbours. The NetNC workflow is summarised in Figure 1 and described in detail below. Two analysis approaches are available (a) node-centric, parameter-free (NetNC-FBT) and (b) edge-centric, with two parameters (NetNC-FTI). Both approaches begin by assigning a *p*-value to each edge (*S_ij_*) from Hypergeometric Mutual Clustering (HMC) [46], described in points one and two, below.

1.A two times two contingency table is derived for each edge *S_ij_* by conditioning on the Boolean connectivity of nodes in *S* to *S_i_* and *S_j_*. Nodes *S_i_* and *S_j_* are not counted in the contingency table.2.Exact hypergeometric *p*-values [46] for enrichment of the nodes in *S* that have edges to the nodes *S_i_* and *S_j_* are calculated using Fisher’s Exact Test from the contingency table. Therefore, a distribution of *p*-values (*H*_1_) is generated for all edges *S_ij_*.3.The NetNC edge-centric analysis setting (NetNC-FTI) employs positive false discovery rate [157] and an iterative minimum cut procedure [158] to derive clusters as follows:
(a)Subgraphs with the same number of nodes as *S* are resampled from *G*, application of steps 1 and 2 to these subgraphs generates an empirical null distribution of neighbourhood clustering *p*-values (*H*_0_). This *H*_0_ accounts for the effect of the sample size and the structure of *G* on the *S_ij_* hypergeometric *p*-values (*p_ij_*). Each NetNC run on TF_ALL in this study resampled 1000 subgraphs to derive *H*_0_.(b)Each edge in *S* is associated with a positive false discovery rate (*q*) estimated over *p_ij_* using *H*_1_ and *H*_0_. The neighbourhood clustering subgraph *C* is induced by edges where the associated *q* ≤ *Q.* Therefore, *Q* is the NetNC-FTI threshold for false discovery rate (*q*).(c)An iterative minimum cut procedure [158] is applied to *C* until all components have density greater than or equal to a threshold *Z*. Edge weights in this procedure are taken as the negative log *p*-values from *H*_1_. Therefore, Z is the threshold for the density of network components output by NetNC-FTI.(d)As described below, thresholds *Q* and *Z* were chosen to optimise the performance of NetNC on the “Functional Target Identification” task using training data taken from KEGG. Connected components with less than three nodes are discarded, in line with common definitions of a “cluster”. Remaining nodes are taken as functionally coherent.4.The node-centric, parameter-free approach (NetNC-FBT) proceeds by calculating degree-normalised node functional coherence scores (NFCS) from *H*_1_, then identifies statistical modes of the NFCS distribution using Gaussian Mixture Modelling (GMM) [159].
(a)The node functional coherence score (NFCS) is calculated by summation of *S_ij_ p*-values in *H*_1_ (*p_ij_*) for fixed *S_i_*, normalised by the *S_i_* degree value in *S* (*di*) (Equation (2)):
(2)$$NFCS_{i}=-\frac{1}{d_{i}}\underset{j}{\sum }\log\left(p_{ij}\right)$$(b)GMM is applied to identify structure in the NFCS distribution. Expectation-maximization fits a mixture of Gaussians to the distribution using independent mean and standard deviation parameters for each Gaussian [159,160]. Models with 1..9 Gaussians are fitted and the final model selected using the Bayesian Information Criterion (BIC).(c)Nodes in high-scoring statistical mode(s) are predicted to be “Functionally Bound Targets” (FBTs) and retained. Firstly, any mode at NFCS < 0.05 is excluded because this typically represents nodes with no edges in *S* (where NFCS = 0). A second step eliminates the lowest scoring mode if >1 mode remains. Very rarely a unimodal model is returned, which may be due to a large non-Gaussian peak at NFCS = 0 confounding model fitting; if necessary, this is addressed by introducing a tiny Gaussian noise component (SD = 0.01) to the NFCS = 0 nodes to produce NFCS_GN0. GMM is performed on NFCS_GN0 and nodes eliminated according to the above procedure on the resulting model. This procedure was developed following manual inspection of results on training data from KEGG pathways with “synthetic neutral target genes” (STNGs) as nodes resampled from *G* (TRAIN-CL).

Therefore, NetNC can be applied to predict functional coherence using either edge-centric or node-centric analysis settings. The NetNC-FTI (edge-centric) approach automatically produces a network, whereas the NetNC-FBT (node-centric) analysis does not output edges; therefore, to generate networks from predicted NetNC-FBT nodes an edge pFDR threshold may be applied, pFDR ≤ 0.1 was selected as the default value. The statistical approach to estimate pFDR and local FDR is described in the sections below.

### 3.3. Estimating Positive False Discovery Rate for Hypergeometric Mutual Clustering p-Values

The following procedure is employed to estimate positive False Discovery Rate (pFDR) [157] in the NetNC-FTI approach (edge-centric). Subgraphs with number of nodes identical to *S* are resampled from *G* to derive a null distribution of HMC *p*-values (*H*_0_) (described above). The resampling approach for pFDR calculation in NetNC-FTI controls for the structure of the network *G*, including degree distribution, (because G is fixed) but does not control for the degree distribution or other network properties of the subgraph *S* induced by the input nodelist (*D*). In scale free and hierarchical networks, degree correlates with clustering coefficient; indeed, this property is typical of biological networks [161]. Part of the rationale for NetNC assumes that differences between the properties of *G* and *S* (for example; degree, clustering coefficient distributions) may enable identification of clusters within *S*. Therefore, it would be undesirable to control for the degree distribution of *S* during the resampling procedure for pFDR calculation because this would also partially control for clustering coefficient. Indeed, clustering coefficient is a node-centric parameter that has similarity with the edge-centric hypergeometric clustering coefficient (HMC) calculation [46] used in the NetNC algorithm to analyse *S*. Hence, the resampling procedure does not model the degree distribution of *S*, although the degree distribution of *G* is controlled for. Positive false discovery rate is estimated over the *p*-values in *H*_1_ (*p_ij_*) according to Storey [157] (Equation (3)):(3)$$pFDR=E\left(\frac{V}{R}\right),R>0.$$

*R* denotes hypotheses (edges) taken as significant, *V* are the number of false positive results (type I error).

NetNC steps through threshold values (*p_α_*) in *p_ij_* estimating *V* using edges in *H*_0_ with *p* ≤ *p_α_*. *H*_0_ represents *Y* resamples, therefore *V* is calculated at each step (Equation (4)):(4)$$V=\frac{H_{0}}{Y},p\leq p_{\alpha }.$$
The *H*_1_
*p*-value distribution is assumed to include both true positives and false positives (FP); *H*_0_ is taken to be representative of the FP present in *H*_1_. This approach has been successfully applied to peptide spectrum matching [162,163]. The value of *R* is estimated by (Equation (5)):(5)$$R=\sum_{p\in H_{1}}\{\begin{array}{c} \;1\;p_{ij}\leq p_{\alpha }\;\\ 0\;otherwise \end{array}.$$
Additionally, there is a requirement for monotonicity (Equation (6)): (6)$$pFDR_{x+1}\geq pFDR_{x}\;,\;p_{x}<p_{x+1}.$$
Equation (6) represents a conservative procedure to prevent inconsistent scaling of pFDR due to sampling effects. For example consider the scaling of pFDR for pFDR_x+1_ at a *p_ij_* value with additional edges from *H*_1_ but where no more resampled edges (i.e., from *H*_0_) were observed in the interval between p_x_ and p_x+1_; before application of Equation (6), the value of pFDR_x+1_ would be lower than pFDR_x_. The approach also requires setting a maximum on estimated pFDR, considering that there may be values of *p_α_* where *R* is less than *V*. We set the maximum to 1, which would correspond to a prediction that all edges at *p_ij_* are FPs. The assumption that *H*_1_ includes false positives is expected to hold in the context of candidate transcription factor target genes and also generally across biomedical data due to the stochastic nature of biological systems [164,165,166,167].

### 3.4. Estimating Local False Discovery Rate from Global False Discovery Rate

We developed an approach to estimate local false discovery rate (lcFDR) [167], being the probability that an object at a threshold (*p_α_*) is a false positive (FP). Our approach takes global pFDR values as basis for lcFDR estimation. In the context of NetNC analysis using the DroFN network, a FP is defined as a gene (node) without a pathway comembership relationship to any other nodes in the nodelist *D*. The most significant pFDR value (pFDR_min_) from NetNC was determined for each node *S_i_* across the edge set *S_ij_*. Therefore, pFDR_min_ is the pFDR value at which node *S_i_* would be included in a thresholded network. We formulated lcFDR for the nodes with pFDR_min_ meeting a given *p_α_* (*k*) as follows (Equation (7)):(7)$$lcFDR_{k}=\frac{\left(\left(n\times pFDR_{k}\right)–\left(\left(n–X\right)\times pFDR_{l}\right)\right)}{X}$$
where *l* denotes the pFDR_min_ closest to and smaller than *k*, and where at least one node has pFDR_min_ ≡ pFDR_l_. Therefore, our approach can be conceptualised as operating on ordered pFDR_min_ values. *n* indicates the nodes in *D* with pFDR_min_ values meeting threshold *k. X* represents the number of nodes at *p_α_* ≡ *k.* The number of FPs for nodes with *p_α_* ≡ *k* (FP*_k_*) is estimated by subtracting the FP for threshold *l* from the FP at threshold *k.* Thus, division of FP*_k_* by *X* gives local false discovery rate bounded by *k* and *l* (Appendix A). If we define the difference between pFDR*_k_* and pFDR*_l_* (Equation (8)):pFDR_Δ_ = pFDR*_k_* − pFDR*_l_*(8)
Substituting pFDR*_k_* for (pFDR*_l_* + pFDR_Δ_) into Equation (7) and then simplifying (Equation (9)): lcFDR*_k_* = ((*n* × pFDR_Δ_)/*X*) + pFDR*_l_*(9)
Equations (7) and (9) do not apply to the node(s) in *D* at the smallest possible value of pFDR_min_ because pFDR_l_ would be undefined; instead, the value of lcFDR*_k_* is calculated as the (global) pFDR_min_ value. Indeed, global FDR and local FDR are equivalent when *H*_1_ consists of objects at a single pFDR_min_ value. Taking the mean lcFDR*_k_* across *D* provided an estimate of neutral binding in the TF_ALL datasets and was calibrated against mean lcFDR values from analysis of “faux” candidate targets resampled from DroFN—where the number of resampled targets was identical to the number of candidate target genes in the TF_ALL dataset analysed. For comparison, we also calibrated mean lcFDR for TF_ALL against values from synthetic data with known %SNTGs (Appendix A). Estimation of the total proportion of neutral binding in ChIP-chip or ChIP-seq data required lcFDR rather than (global) pFDR and, for example, accounts for the shape of the *H*_1_ distribution. In the context of NetNC analysis of TF_ALL, mean lcFDR may be interpreted as the probability that any candidate target gene is neutrally bound in the dataset analysed; therefore, providing estimation of the total neutral binding proportion. Computer code for calculation of lcFDR is provided within the NetNC distribution. Estimates of SNTGs by the NetNC-FBT approach were not taken forward due to large 95% CI values (Appendix A).

### 3.5. Median Difference and Correlation between Estimates of Functional Binding from NetNC Functional Target Identification and Local False Discovery Rate 

Candidate target genes that passed NetNC-FTI thresholds were considered functional targets (FTI_FT). The proportion of FTI_FT genes was compared to the proportion of functional binding estimated by lcFDR (lcFDR_FB, Figure 3A). The modulus of the difference between FTI_FT and lcFDR_FB for each dataset gave a distribution of differences in predicted functional binding and the median of this distribution was quoted in the text above.

### 3.6. NetNC Benchmarking Data

There are significant challenges in separating functional TF targets from neutral binding in experimentally determined candidate TF target gene lists [1,2,5]. For example, it is not straightforward to define neutral TF binding by examining target genes that do not change steady-state expression upon knockout of the TF in question: TF knockout might not affect expression of *bona fide* TF targets where an additional TF (or TFs) are partially redundant with the lost TF [2,5]; and loss of TF binding may alter gene expression dynamics, such as a change in oscillation or stochasticity, without changing steady-state gene expression measured across a large population of cells [164,165]. Gene expression changes following TF knockout can also incorrectly propose “functional” targets that change expression via an indirect mechanism. For example, manipulation of gene expression by knockout or overexpression causes systemic changes in gene expression, including regulation by feedback loops with complicated logical outputs [6]. Therefore, we developed synthetic benchmark data for the purpose of development and evaluation of NetNC—where biologically coherent genes were taken to represent functional TF targets, and randomly sampled genes taken to represent neutral binding. We consider the relevance of our synthetic benchmark data to the nine TF_ALL datasets in the section below; firstly we outline the construction of the synthetic benchmark. Gold standard data for NetNC benchmarking and parameterisation took pathways from KEGG to represent biologically coherent gene groups (v62, downloaded 13/6/12) [41]. Training data were selected as seven pathways (TRAIN-CL, 184 genes) and a further eight pathways were selected as a blind test dataset (TEST-CL, 186 genes) summarised in Appendix A. For both TRAIN-CL and TEST-CL, pathways were selected to be disjoint and to cover a range of different biological functions. However, pathways with shared biology were present within each group; for example, TRAIN-CL included the pathways dme04330 “Notch signaling” and dme04914 “Progesterone-mediated oocyte maturation”, which are related by notch involvement in oogenesis [168,169]. TEST-CL also included the related pathways dme04745 “Phototransduction” and dme00600 “Sphingolipid metabolism”, for example where ceramide kinase regulates photoreceptor homeostasis [170,171,172].

Gold standard datasets were also developed in order to investigate the effect of dataset size and noise on NetNC performance. The inclusion of noise as resampled network nodes into the gold-standard data was taken to model neutral TF binding [1,7] and matches expectations on data taken from biological systems in general [164,167]. Therefore, gold standard datasets were generated by combining TRAIN-CL with nodes resampled from the network (*G*). The final proportion of resampled nodes (Synthetic Neutral Target Genes, SNTGs) ranged from 5% through to 80% in 5% increments. SNTGs were drawn by uniform resampling from the DroFN network using the rand() function in perl excluding the genes in TRAIN-CL. Since we expected variability in the network proximity of SNTGs to pathway nodes, 100 resampled datasets were generated per %SNTG increment. Additional gold-standard datasets were generated by taking five subsets of TRAIN-CL, from three through seven pathways. Resampling was applied for these datasets as described above to generate node lists representing five pathway sets in TRAIN-CL by sixteen %SNTG levels by l00 repeats (TRAIN_CL_ALL, 8000 node lists). A similar procedure was applied to TEST-CL, taking from three through eight pathways to generate data representing six pathway subsets by sixteen noise levels by 100 repeats (TEST-CL_ALL, 9600 node lists). Data based on eight pathways (TEST-CL_8PW, 1600 node lists) were used for calibration of lcFDR estimates. Preliminary training and testing against the MCL algorithm [32] utilised a single subsample for 10%, 25%, 50%, and 75% SNTGs (TRAIN-CL-SR, TEST-CL-SR).

### 3.7. Comparison of Synthetic NetNC Benchmark to Experimentally Determined TF Binding Data

Transcription factors act to coordinately regulate multiple functionally related targets [20,21,22]. Accordingly, we reasoned that biological pathways could be taken as synthetic functional TF targets. We modelled neutral binding by random resampling across all of the genes represented in the DroFN network, with the rationale that the creation and disappearance of neutral binding sites would not be driven by evolutionary selection. In order to further explore the correspondence between the synthetic benchmark and the TF binding datasets analysed (TF_ALL), we compared the global clustering coefficient (CC) of the network induced in DroFN by the nine TF_ALL datasets and the synthetic benchmark (TEST-CL_ALL). CC provides a single graph-theoretic measure of clustering in each dataset, which is a key property used by NetNC to identify functional TF targets. Therefore, CC is an appropriate measure to use for comparison of the synthetic and experimentally determined datasets. Accordingly, we developed linear models using TEST-CL_ALL in order to predict the CC of the biological TF_ALL data. Each dataset in TF_ALL had been assigned to a proportion of neutral binding, matched to a %SNTG value in the synthetic benchmark (Appendix A). For model training, we took the mean CC across the 100 repeats per %SNTG level for the six different dataset sizes in the synthetic data (from three up to eight pathways in each). This generated six values of CC and six values for the number of nodes (#nodes) in the dataset per %SNTG, which were used for the model:CC = intercept + (coefficient × log(#nodes)).(10)

As an illustrative example, the sna_2–3h_union dataset was matched to the benchmark dataset with 75% SNTGs. Therefore, a model was fitted using the TEST-CL_75% dataset with the six values for the number of nodes (for each of the six sub-datasets from three to eight pathways) and their six corresponding CC values. The model based on TEST-CL_75% was used to predict CC for sna_2–3h_union; the predicted CC value was compared to the CC calculated for the graph induced by sna_2–3h_union in DroFN. The fitted models therefore accounted for the expected influence of dataset size on the CC of the induced subnetwork. The “glm” function in R was used for model fitting. The regularised fit, determined by Akaike Information Criterion, was always superior for models where the logarithm of the number of nodes was taken, rather than taking raw values. The calculated CC values were for subnetworks induced in DroFN by the relevant TF binding dataset and included disconnected nodes.

### 3.8. NetNC-FTI Parameter Optimisation

NetNC-FTI analysed the TRAIN-CL_ALL datasets across a range of FDR (*Q*) and density (*Z*) threshold values. Performance was benchmarked on the Functional Target Identification (FTI) task which assessed the recovery of biological pathways and exclusion of SNTGs. Matthews correlation coefficient (MCC) was computed as a function of NetNC-FTI parameters (Q, Z). MCC is attractive because it is captures predictive power in both the positive and negative classes. FTI was a binary classification task for discrimination of pathway nodes from noise, therefore all pathway nodes were taken as positives and SNTGs were negatives for the FTI MCC calculation. The NetNC-FTI approach therefore tests discrimination of pathway nodes from SNTGs, which is particularly relevant to identification of functionally coherent candidate TF targets from ChIP-chip or ChIP-seq peaks.

Parameter selection for NetNC-FTI analysed MCC values for the 100 SNTG resamples across five pathway subsets by sixteen SNTG levels in TRAIN-CL_ALL over the Q, Z values examined, respectively ranging from up to 10^−7^ to 0.8 and from up to 0.05 to 0.9. Data used for optimisation of NetNC-FTI parameters (Q, Z) are available from the BioStudies database (www.ebi.ac.uk/biostudies/studies/S-BSST460) and contour plots showing mean MCC across Q, Z values per %SNTG are provided in Appendix A. A “SNTG specified” parameter set was developed for situations where an estimate of the input data noise component is available, for example from the NetNC-FBT approach. In this parameterisation, for each of the 16 datasets with different proportions of SNTG (5%.. 80%), MCC values were normalized across the five pathway subsets of TRAIN-CL (from three through seven pathways), by setting the maximum MCC value to 1 and scaling all other MCC values accordingly. The normalised MCC values < 0.75 were set to zero and then a mean value was calculated for each %SNTG value across five pathway subsets by 100 resamples in TRAIN-CL_ALL (500 datasets per noise proportion). This approach therefore only included parameter values corresponding to MCC performance ≥ 75% of the maximum across the five TRAIN-CL pathway subsets. The high performing regions of these “summary” contour plots sometimes had narrow projections or small fragments, which could lead to parameter estimates that do not generalise well on unseen data. Therefore, parameter values were selected as the point at the centre of the largest circle (in (Q, Z) space) completely contained in a region where the normalised MCC value was ≥0.95. This procedure yielded a parameter map: (SNTG Estimate) → (Q, Z), given in Appendix A. Parameters were also determined for analysis without any prior information about the %SNTG in the input data. For this purpose, a contour plot was produced to represent the proportion of datasets where NetNC-FTI performed better than 75% of the maximum performance across TRAIN-CL_ALL for the FTI task in the Q, Z parameter space. The maximum circle approach described above was applied to the contour plot in order to derive “robust” parameter values (Q, Z), which were respectively 0.120, 0.306 (NetNC-FTI).

### 3.9. Performance on Blind Test Data

We compared NetNC-FTI and NetNC-FBT against leading methods, HC-PIN [33] and MCL [32] on blind test data (Figure 2, Appendix A). HC-PIN was obtained from the developers and is currently available within the cytocluster Cytoscape app (https://apps.cytoscape.org/apps/cytocluster); MCL is available from https://micans.org. Previous work that evaluated nine clustering algorithms, including MCL, found that HC-PIN had strong performance in functional module identification and was robust against false positives [33]; therefore HC-PIN was selected for extensive comparison against NetNC. Input, output and performance summary files for HC-PIN on TEST-CL are available from the BioStudies database (per datapoint, *n* = 100 for NetNC, *n* = 99 for HC-PIN). HC-PIN was run on the weighted graphs induced in DroFN by TEST-CL with default parameters (lambda = 1.0, threshold size = 3). MCL clusters in DroFN significantly enriched for query nodes from TEST-CL-SR were identified by resampling to generate a null distribution [152]. Clusters with *q* < 0.05 were taken as significant. MCL performance was optimised for the functional target identification (FTI) task over the TRAIN-CL-SR datasets for MCL inflation values from 2 to 5 incrementing by 0.2. The best-performing MCL inflation value overall was 3.6 (Appendix A). Comparison to NEST [52] and baseline node degree was performed on TEST-CL-SR (Appendix A). NEST required expression values, therefore a uniform expression value was added to the NEST input for all TEST-CL-SR nodes. The NEST output included genes that were not present in the input data from TEST-CL-SR and these additional genes were removed in order to produce the “Filtered NEST” dataset. The NEST scores or node degree were analysed separately against the labelling of TEST-CL-SR nodes as KEGG pathways (positives) or SNTGs (negatives), enabling calculation of area under the Receiver Operator Characteristic curve for each method examined (Appendix A).

### 3.10. Subsampling of Transcription Factor Binding Datasets and Statistical Testing

Robustness of NetNC performance was studied by taking 95%, 80%, and 50% resamples from nine public transcription factor binding datasets, summarised above and described previously in detail [8,10,22,53,54]. A hundred subsamples of each of these datasets were taken at rates of 95%, 80%, and 50%, thereby producing a total of 2700 datasets (TF_SAMPL). NetNC-FTI results across TF_SAMPL were used as input for calculation of median and 95% confidence intervals for the edge and gene overlap per subsampling rate for each transcription factor dataset analysed. The NetNC resampling parameter (Y) was set at 100, the default value. The edge overlap was calculated as the proportion of edges returned by NetNC-FTI for the subsampled dataset that were also present in NetNC-FTI results for the full dataset (i.e., at 100%). Therefore, nine values for median overlap and 95% CI were produced per subsampling rate for both edge and gene overlap, corresponding to the nine transcription factor binding datasets (Appendix A). The average (median) value of these nine median overlap values, and of the 95% CI, was calculated per subsampling rate; these average values are quoted in Appendix A.

False discovery rate (FDR) correction of *p*-values was applied where appropriate and is indicated in this manuscript by the commonly used notation “*q*” Benjamini–Hochberg correction was applied [173] unless otherwise specified in the text. Calculation of pFDR and local FDR values by NetNC is described in the sections above.

### 3.11. Transcription Factor Binding and Notch Modifier Datasets

We analysed public Chromatin Immunoprecipitation (ChIP) data for the transcription factors *twist* and *snail* in early *Drosophila melanogaster* embryos. These datasets were derived using ChIP followed by microarray (ChIP-chip) [22,53,54] and ChIP followed by solexa pyrosequencing (ChIP-seq) [8]. Additionally “highly occupied target” regions, reflecting multiple and complex transcription factor occupancy profiles, were obtained from ModEncode [10]. Nine datasets were analysed in total (TF_ALL) and are summarised below.

The “union” datasets (WT embryos 2–3 h, mostly late stage four or early stage five) combined ChIP-chip peaks significant at 1% FDR for two different antibodies targeted at the same TF and these were assigned to the closest transcribed gene according to RNA Polymerase II binding data [22]. Additionally, where the closest transcribed gene was absent from the DroFN network then the nearest gene was included if it was contained in DroFN. This approach generated the datasets sna_2–3h_union (1158 genes) and twi_2–3h_union (1848 genes). The union of peaks derived from two separate antibodies maximised sensitivity and may have reduced potential false negatives arising from epitope steric occlusion. For the “Toll^10b^” datasets, significant peaks with at least two-fold enrichment for Twist or Snail binding were taken from ChIP-chip data on Toll^10b^ mutant embryos (2–4h), which had constitutively activated Toll receptor [54,174]; mapping to DroFN generated the datasets twi_2–4h_Toll^10b^ (1238 genes), sna_2–4h_Toll^10b^ (1488 genes). Toll^10b^ embryos had high expression of Snail and Twist, which drove all cells to mesodermal fate trajectories [54]. The two-fold enrichment threshold selected for this study reflects “weak” binding, although was expected to include functional TF targets [9]. Therefore, the candidate target genes for twi_2–4h_Toll^10b^ and sna_2–4h_Toll^10b^ were expected to contain a significant proportion of false positives. The Highly Occupied Target dataset included 38,562 regions, of which 1855 had complexity score ≥8 and had been mapped to 1648 FlyBase genes according to the nearest transcription start site [10]; 677 of these genes were matched to a DroFN node (HOT). The “HighConf” data took Twist ChIP-seq binding peaks in WT embryos (1–3 h) that had been reported to be “high confidence” assignments; high confidence filtering was based on overlap with ChIP-chip regions, identification by two peak-calling algorithms and calibration against peak intensities for known Twist targets, corresponding to 832 genes [8] of which 755 were mapped to FlyBase. A total of 664 of these genes were found in DroFN (twi_1–3h_hiConf) and represented the most stringent approach to peak calling of all the nine TF_ALL datasets. The intersection of ChIP-chip binding for two different Twist antibodies in WT embryos spanning two time periods (2–4h and 4–6h) identified a total of 1842 target genes [53] of which 1444 mapped to DroFN (Intersect_ALL). Subsets of Intersect_ALL identified regions bound only at 2–4 h (twi_2–4h_intersect, 801 genes), or only at 4–6 h (twi_4–6h_intersect, 818 genes), or “continuously bound” regions identified at both 2–4 and 4–6 h (twi_2–6h_intersect, 615 genes). Assigned gene targets may belong to more than one subset of Intersect_ALL because time-restricted binding was assessed for putative enhancer regions prior to gene mapping; overlap of the Intersect_ALL subsets ranged between 30.2% and 55.4%. The Intersect_ALL datasets therefore enabled assessment of functional enhancer binding according to occupancy at differing time intervals and also to examine the effect of intersecting ChIPs for two different antibodies upon the proportion of predicted functional targets recovered.

Seven of the nine TF_ALL datasets included developmental time periods encompassing stage four (syncytial blastoderm, 80–130 min), cellularisation of the blastoderm (stage five, 130–170 min) and initiation of gastrulation (stage 6, 170–180 min) [8,22,53,54,175]. The datasets twi_2–4h_intersect, sna_2–4h_intersect, twi_2–4h_Toll^10b^ and sna_2–4h_Toll^10b^ additionally included initial germ band elongation (stage seven, 180–190 min) [53,54,175]; twi_2–4h_Toll^10b^ and sna_2–4h_Toll^10b^ may have also included stages eight (190–220 min) and nine (220–260 min) [54,175]. Twi_2–4h_intersect and sna_2–4h_intersect were tightly staged between stages 5–7 [53]. Additional to stages four, five and six, twi_1–3h_hiConf may have included the latter part of stage two (preblastoderm, 25–65 min) and stage three (pole bud formation, 65–80 min) [175]. The twi_4–6h_intersect dataset was restricted to stages eight to nine which included germ band elongation and segmentation of neuroblasts [53,175]. Therefore, there were differences in the biological material used across TF_ALL.

The Notch signalling modifiers analysed in this study were selected based on identification in at least two of the screens reported in [64]. Networks were annotated using GO and FlyBase [31,39,176,177].

### 3.12. Breast Cancer Transcriptome Datasets and Molecular Subtypes

Primary breast tumour gene expression data were downloaded from NCBI GEO (GSE12276, GSE21653, GSE3744, GSE5460, GSE2109, GSE1561, GSE17907, GSE2990, GSE7390, GSE11121, GSE16716, GSE2034, GSE1456, GSE6532, GSE3494, and GSE68892 (formerly geral-00143 from caBIG)). All datasets were Affymetrix U133A/plus 2 chips and were summarised with Ensembl alternative CDF [178]. RMA normalization [179] and ComBat batch correction [180] were applied to remove dataset-specific bias as previously described [110,181]. Intrinsic molecular subtypes were assigned based upon the highest correlation to Sorlie centroids [104], applied to each dataset separately. Centred average linkage clustering was performed using the Cluster and TreeView programs [182]. Centroids were calculated for each gene based upon the mean expression across each of the Sorlie intrinsic subtypes [104]. These expression values were squared to consider up and down regulated genes in a single analysis. Orthology to the DroFN network was defined using Inparanoid [183]. Differential expression was calculated by t-test comparing normalised (unsquared) expression values in normal-like and basal-like tumours with false discovery rate correction [173].

### 3.13. Invasion Assays for Validation of Genes Selected from NetNC Results

MCF-7 Tet-On cells were purchased from Clontech and maintained as previously described [184]. To analyse the ability of transfected MCF7 breast cancer cells to degrade and invade surrounding extracellular matrix, we performed an invasion assay using the CytoSelect™ 24-Well Cell Adhesion Assay kit. This transwell invasion assay allows the cells to invade through a matrigel barrier utilising basement membrane-coated inserts according to the manufacturer’s protocol. Briefly, MCF7 cells transfected with the constructs (Doxycycline-inducible *SNAI1* cDNA or *SNAI1 shRNA* with or without candidate gene cDNA) were suspended in serum-free medium. *SNAI1* cDNA or *SNAI1 shRNA* were cloned in our doxycyline-inducible pGoldiLox plasmid (pGoldilox-Tet-ON for cDNA and pGolidlox-tTS for shRNA expression) using validated shRNAs against *SNAI1* (NM_005985 at position 150 of the transcript [184]). pGoldilox has been used previously to induce and knock down the expression of *Ets* genes [185]. Following overnight incubation, the cells were seeded at 3.0 × 10^5^ cells/well in the upper chamber and incubated with medium containing serum with or without doxycyline in the lower chamber for 48 h. Concurrently, 10^6^ cells were treated in the same manner and grown in a six well plate to confirm over-expression and knockdown. mRNA was extracted from these cells and quantitative real-time PCR (RT-qPCR) was performed as previously described [186]; please see Data File S2 for gene primers. The knockdown efficiency for Snai1 was >81% (5.4-fold knockdown), Snai1 induction produced 2.0-fold overexpression. The transwell invasion assay evaluated the ratio of CyQuant dye signal at 480/520 nm in a plate reader of cells from the two wells and therefore controlled for potential proliferation effects associated with ectopic expression. We used empty vector (mCherry) and scrambled shRNA as controls and to control for the non-specific signal. At least three experimental replicates were performed for each reading.

### 3.14. Data and Software Availability

NetNC is available at https://github.com/overton-group/NetNC. The following data are available from the European Bioinformatics Institute BioStudies database (https://www.ebi.ac.uk/biostudies/studies/S-BSST460): The DroFN network; all gold standard datasets; HCPIN input, output and performance summary files. The all-vs-all connectivity matrix before application of the DroFN edge weight threshold is available upon request.

## 4. Conclusions

We developed and validated the novel NetNC algorithm for identification of biologically coherent transcription factor (TF) target genes, and a comprehensive *D. melanogaster* functional gene network (DroFN). While NetNC was developed for functional TF target discovery, the approach may be widely useful for recovery of functionally coherent nodes in noisy data, for example in analysis of differential gene expression or CRISPR screen data. The network-based statistical framework in NetNC is applicable to single sample datasets and includes a novel method for estimation of local false discovery rate (FDR) from global FDR values. Analysis of Snail, Twist and modENCODE highly occupied target (HOT) regions found from 50% to 95% of candidate target genes were neutrally bound across the nine datasets analysed (TF_ALL). Correlation of the predicted neutral binding proportion with experimental and analytical factors across TF_ALL suggested consideration of strategies to enrich for functional TF targets. Datasets representing > 1 time period or that were derived from multiple TFs (HOT regions) had a relatively high proportion of functional binding, aligning with the emerging picture of widespread combinatorial control involving TF–TF interactions, cooperativity and TF redundancy [2,5,57,58,59]. The NetNC functional target networks provide a map of genome-scale regulation by Snail and Twist in early *D. melanogaster* embryogenesis. Each of the networks for the nine TF_ALL datasets was significantly enriched in Notch signalling modifiers, and we predicted genes involved in signalling cross-talk—where Snail and Twist may act to control the pleiotropic consequences of Notch activation. Eleven biological functions were annotated to at least four of the nine TF_ALL networks, including developmental regulation, chromatin organisation and mushroom body development. Predicted Snail and Twist regulation of chromatin structure, including PRC1 core components and other gene silencing factors, provides evidence for the action of EMT TFs in controlling the expression of their own coregulators. Unsupervised clustering with orthologues of the NetNC functional targets stratified 2999 breast cancer transcriptomes into the five intrinsic subtypes [104]; demonstrating that the regulation by Snail and Twist in fly mesoderm development captures important features of breast cancer biology. We identified breast cancer subtype-specific genes and network modules. Results in the basal-like subtype suggest a role for *HECA* in an EMT-like gene expression programme, and predict orthologous Snail, Twist functional targets that may control the consequences of *Notch* activation through chromatin remodelling. Our integrative analysis revealed subtype-specific genes that may prove useful for precision medicine. For example, potentially informing development of companion diagnostics or combination therapies targeting the notch pathway in basal-like tumours. We validated predicted roles in invasion for four NetNC functional targets in a breast cancer model, supporting our approach.

## Figures and Tables

**Figure 1 cancers-12-02823-f001:**
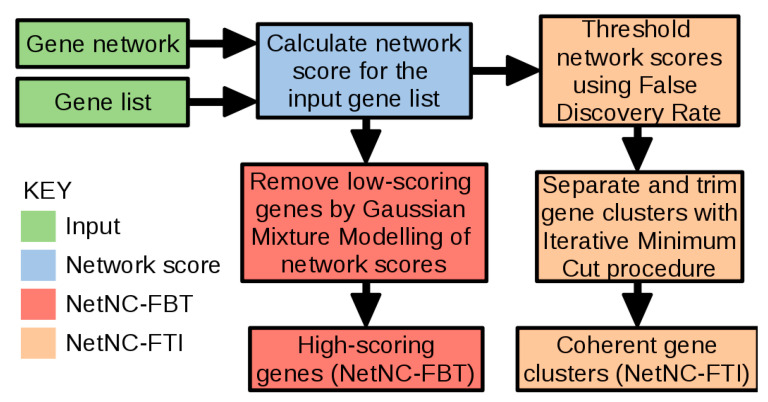
Overview of the NetNC algorithm. A gene network and a gene list are required by NetNC as input (green). The current study analyses candidate TF targets; however, NetNC could be applied to any gene list and network. NetNC calculates a network score (blue) using Hypergeometric Mutual Clustering (HMC) for each gene pair in the input gene list according to connections in the network. Two analysis settings are (a) NetNC-FBT (red), where Gaussian Mixture Modelling identifies high-scoring genes; (b) NetNC-FTI (orange), which produces coherent gene clusters by thresholding network scores according to False Discovery Rate followed by Iterative Minimum Cut.

**Figure 2 cancers-12-02823-f002:**
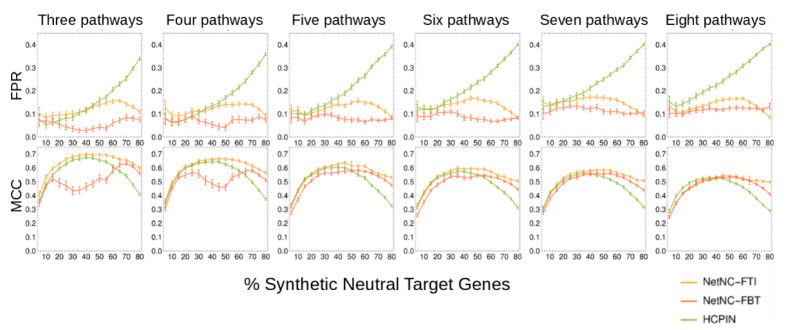
Evaluation of NetNC and HC-PIN on blind test data. Performance values reflect discrimination of KEGG pathway nodes from Synthetic Neutral Target Genes (STNGs), shown for NetNC-FTI (orange), NetNC-FBT (red) and HC-PIN (green). False positive rate (FPR, top row) and Matthews Correlation Coefficient (MCC, bottom row) values are given. Data shown represents analysis of TEST-CL_ALL, which included subsets of three to eight pathways, shown in columns, and sixteen %STNG values were analysed (5% to 80%, *x*-axis). NetNC performed best on the data examined with typically lower false positive rate and higher MCC values. Error bars reflect 95% confidence intervals calculated from quantiles of SNTG resamples. Also see Appendix A.

**Figure 3 cancers-12-02823-f003:**
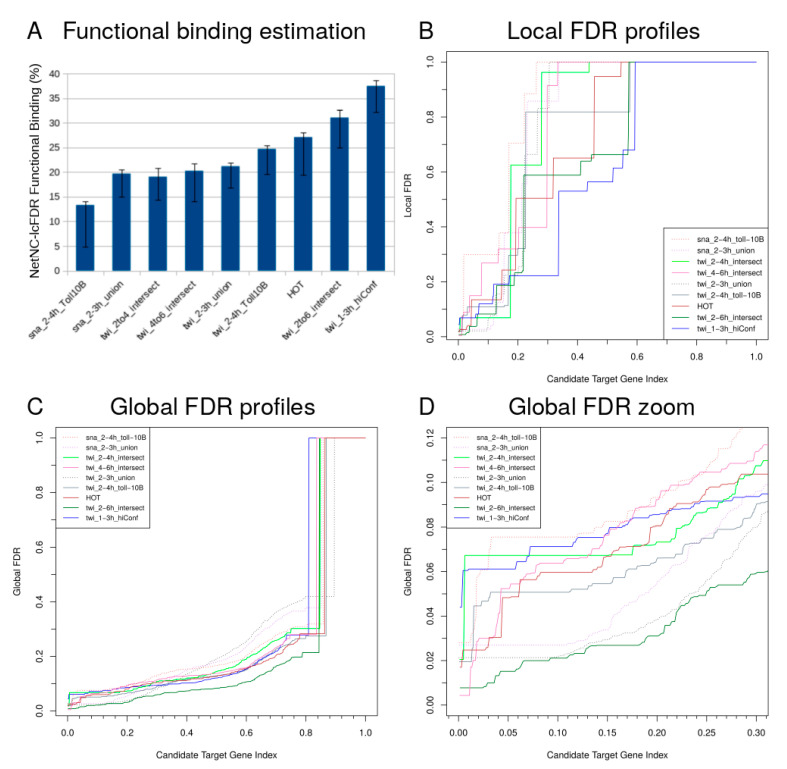
Functional transcription factor binding and false discovery rate (FDR) profiles. (**A**) Estimation of total functional binding. Median values are shown for NetNC-lcFDR, calibrated against resampled genes. Error bars represent 95% CI calculated using quantiles of results for the resampled datasets, which gave predicted functional binding ranging from 5% (sna_2–3h lower CI) to 39% (twi_1–3h_hiConf upper CI) across TF_ALL. Calibration based on synthetic data resulted in slightly higher functional binding estimates, up to 50% (Appendix A). (**B**–**D**) Line type and colour indicates dataset identity (see key). Candidate target gene index values were normalised from zero to one, in order to enable comparison across the TF_ALL datasets.

**Figure 4 cancers-12-02823-f004:**
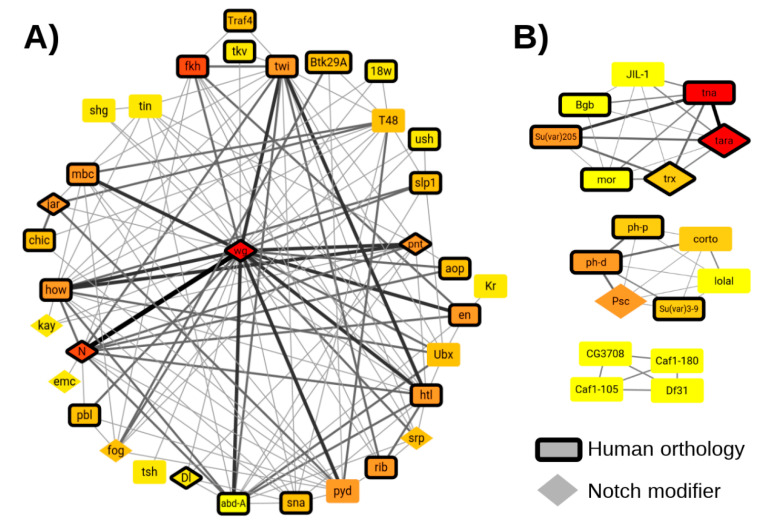
Clusters of developmental regulation and chromatin organisation genes identified by NetNC in multiple TF_ALL datasets. These clusters visualise the combined NetNC-FTI results across the nine TF_ALL datasets; node fill colour, edge width and edge colour indicate frequency of occurrence. Notch modifiers (diamonds) were reported in at least two screens and genes with InParanoid human orthology are shown with black borders. NetNC-FTI clusters for the individual TF_ALL datasets are shown in Appendix A and are available in Cytoscape format (Data File S1). (**A**) Developmental regulation cluster genes in at least five (yellow), up to nine (red) datasets. Edges shown were in ≥5/9 datasets, up to a maximum of 8/9 for *N* and *wg*. Thirteen genes (DRC_13) were present in ≥7/9 datasets, including *wg* (9/9) which had highest degree, *N* (8/9) and *fkh* (8/9). (**B**) Genes in chromatin organisation clusters from two (yellow) up to six (red) TF_ALL datasets. The three clusters were associated with trithorax-group (top), polycomb group (middle), and chromatin assembly factors (bottom). These results predict components regulated by Snail and Twist in establishing the chromatin blueprint for mesodermal lineages.

**Figure 5 cancers-12-02823-f005:**
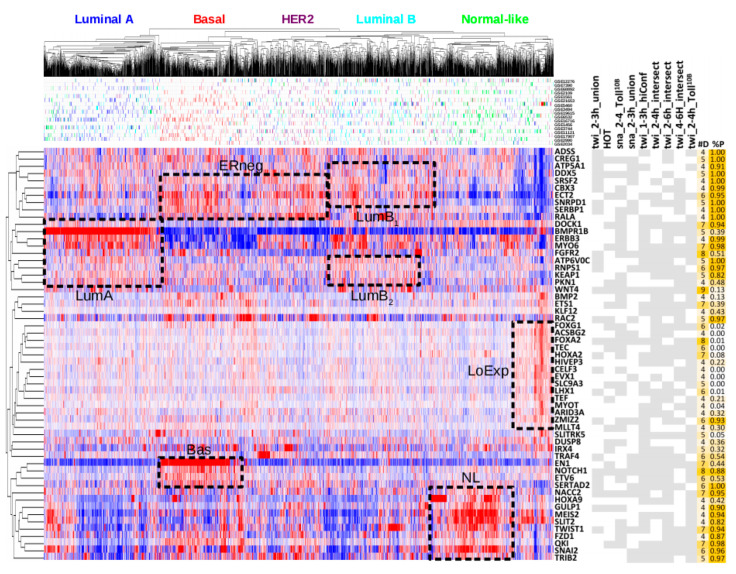
Predicted functional transcription factor targets capture human breast cancer biology. The heatmap shows gene expression in 2999 primary breast tumours for ORTHO-57 genes (red = high, white = mean, blue = low). The mosaic above the heatmap indicates intrinsic molecular subtype: Luminal A (blue), basal-like (red), HER2-overexpressing (purple), luminal B (light blue), and normal-like (green). Annotated heatmap features (black dashed lines) identified genes upregulated in one or more intrinsic subtype; “Bas” (basal-like), “NL”(normal-like), “ERneg” (basal-like and HER2-overexpressing), “LumB_1_”(luminal B), “LumB_2_”(luminal B), “LumA” (luminal A), and “LoExp” (low expression). The table to the right of the heatmap indicates inclusion (grey) or absence (white) across TF_ALL; the number of datasets where the gene was identified by NetNC-FTI (#D) and the percentage of present calls across the 2999 tumours (%*p*) are shown. The LoExp feature corresponded overwhelmingly to genes with low %*p* values and to samples from a single study [121]. Some genes were annotated to more than one feature and reciprocal patterns of gene expression were found. For example, *BMPR1B*, *ERBB3*, and *MYO6* were strongly upregulated in feature LumA but downregulated in basal-like and *HER2*-overexpressing cancers. Unexpectedly, feature NL (normal-like) had high expression of canonical EMT drivers, including *SNAI2*, *TWIST,* and *QKI*. Some of the EMT genes in feature NL were upregulated in many basal-like tumours, while genes in feature Bas (*NOTCH*, *SERTAD2*) had relatively high expression in normal-like tumours. Also see Appendix A.

**Figure 6 cancers-12-02823-f006:**
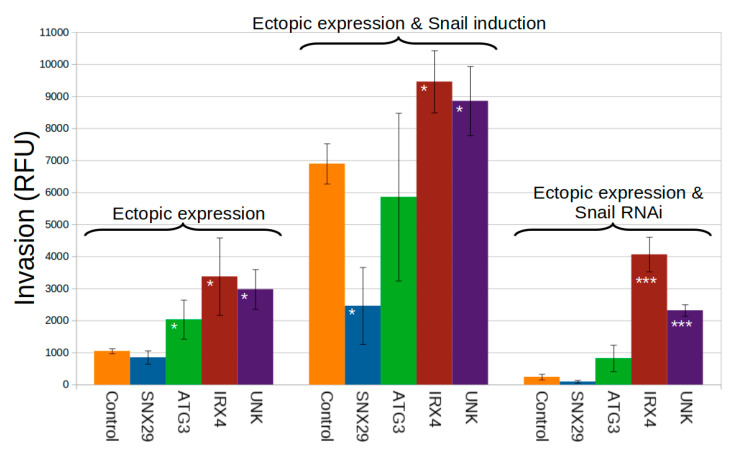
Validation of candidate invasion genes in breast cancer cells. The fluorescence signal from invasive MCF7 cells is shown. Induction of every gene examined significantly changed invasion in least one of three conditions: (a) Ectopic expression; (b) ectopic expression and *SNAI1* induction; (c) ectopic expression with shRNA knockdown of *SNAI1*. *SNX29* (blue) had reduced invasion compared with the *SNAI1* induction control (orange); *UNK* (purple) and *IRX4* (dark red) had increased invasion in all three conditions examined; *ATG3* (green) had higher invasion at background levels of *SNAI1* (without induction or knockdown). Mean values are shown, error bars indicate 95% CI, *n* = 3; * *q* < 0.05; *** *q* < 5.0 × 10^−4^.

**Table 1 cancers-12-02823-t001:** Predicted functional binding for Snail, Twist and Highly Occupied Target (HOT) candidate target genes. The developmental time periods correspond to the following developmental stages: 2–4 h stages 4–9 (except 2–4 h_intersect datasets which were stages 5–7 [53]); 2–3 h stages 4–6; 1–3 h stages 2–6; 4–6 h stages 8–9 [53]; 0–12 h stages 1–15. Also see Appendix A.

Dataset				Predicted Functional Targets ^‡^	
Name	Developmental Time Period(s)	Total Candidate Target Genes *	Candidate Target Genes in DroFN	NetNC-FTI	NetNC-lcFDR (95% CI)
twi_1–3h_hiConf	1–3 h	755	664	202 (30%)	37% (32–39%)
twi_2–6h_intersect	2–4 h and 4–6 h	743	615	241 (39%)	31% (25–33%)
twi_2–4h_intersect	2–4 h only (not 4–6 h)	1028	801	182 (23%)	19% (14–21%)
twi_4–6h_intersect	4–6 h only (not 2–4 h)	1026	818	126 (15%)	20% (14–22%)
HOT	0–12 h ^+^	1648	677	174 (26%)	27% (19–28%)
twi_2–3h_union	2–3 h	2285	1848	424 (23%)	21% (17–22%)
sna_2–3h_union	2–3 h	1424	1158	226 (20%)	20% (15–21%)
twi_2–4h_Toll^10b^	2–4 h	1578	1238	279 (23%)	25% (20–25%)
sna_2–4h_Toll^10b^	2–4 h	1822	1488	211 (14%)	13% (5–14%)

* mapped to FlyBase; **^‡^** for candidate target genes in DroFN; ^+^ multiple time periods and 41 different transcription factors (TFs).

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
