# Peer review of "Functional Transcription Factor Target Networks Illuminate Control of Epithelial Remodelling"

_cancers, 2020, doi:10.3390/cancers12102823_

Round 1
Reviewer 1 Report
Gene regulatory networks play a key role in understanding development and cancer progression. Those important regulatory pathways are organized in a modular and conserved pattern across species. Here, the authors present a new software called NetNC along with a new D.melanogaster network to leverage the membership information for prioritizing a given gene list from ChIP-seq experiment or other high throughput techniques. The prioritized gene set by NetNC analysis of D.melanogaster ChIP-seq could be possibly applied as biomarkers for classifying the breast cancer subtypes and invasion levels. NetNC has the potential as a useful tool to help experimentalists to priories an interesting gene list. However, there are some major and minor issues that arise during the review. Here is the list of them.
Major comments:
- In section 2.1, the NetNC has a new way to construct the integrated network from multiple sources like KEGG, GO terms and KEGG pathways, is it a flexible option to customize any input network (e.g. STRING database directly, or Reactome) for the NetNC following analysis? Is the network integrative construction applicable to other model species? Is the replacement of DroFN with GeneMania, DroID network significant the model performance for both simulation and real datasets? Otherwise, constraining the network input from fly would largely limit the general application of the software.
- In Figure 2, are HC-PIN and MCL methods implemented by the author? No software links are found in the methods or the reference. NetNC also should be compared to more specific bioinformatics software like NEST (https://genomebiology.biomedcentral.com/articles/10.1186/s13059-015-0808-9) and top three methods from the DREAM challenge, and the baseline gene node degree.
- FBT method seems to be the best method with highest MCC and lowest FPR with 80% of STNR, however, in the subsequent analysis of real datasets, the author only use the FTI instead for all analysis. The reason behind this is shortly mentioned in the method that FBT has a higher variance, why does the node-centric method have higher variance than edge-centric one? It is still not clear why not applied both of them to Table 1 and the following analysis. One tip is to combine the two ranking scores together to provide a potentially better integrative score for benchmarking.
- In section 2.3, authors use a binary way to predict functional target genes, which is a poor method, exponential decay-based function (ChIP-Seq of transcription factors predicts absolute and differential gene expression in embryonic stem cells) or ClosestGene (https://journals.plos.org/ploscompbiol/article?id=10.1371/journal.pcbi.1003342#pcbi.1003342-Ouyang1) method are recommended to predict ChIP-seq targets. In Table 1, the lcFDR should be mentioned briefly before the section, what's the purpose of these FDR, may be inserted to line 107. It will be better to have the corresponding gene differential expression data in Table 1 to have better functional predicted targets and neutral genes.
- In section 2.5, why the authors does not choose a high-quality cohort from TCGA? Also, Basal and Her2 subtypes are clustered together which looks good, but why Luminal A do not cluster well with Luminal B? What does the TCGA survival curve look like for those candidate genes in line 381-382? Are these NetNC predicted genes the differentially expressed genes in the cohort?
Minor comments:
- In line 92, is TEST-NET a gold standard to benchmark network quality? A brief introduction should be described in the main text.
- How could NetNC be applied to single-subject datasets in line 108? Could the authors give a demo to illustrate the point?
- The 9 datasets in line 128 should be first briefly described previous published biological stories and then present the analysis results. Also, ar these 9 datasets the same as Table 1?
- The gold standard data using SNTG is based on uniformly random sampling from the network, which is kind of too simple. Does the author need to consider sampling genes with similar degrees, such as the network rewiring strategy in NEST?
- When input a gene list, does NetNC consider the gene expression or associated ChIP-seq signal or CRISPR hits to reweight the score of a gene node or edges?
- In Figure 3, the A panel is duplicated as in Table 1, does the x-axis of B-D panel mean the normalized gene ranks?
- line 227-231, could authors provide the absolute correlation of the ChIP-seq regulatory potential against the Net-FTI score? How does the peak score calculated, is it normalized by sequence depth for comparison?
- In line 286, is the combination of chromatin organization clusters at the gene set level or network level.
- In Figure 5, Could the mosaic above the heatmap combined into one color annotation by the proportion of certain cancer subtypes? Also, the dash box line color can be changed to the same font color in the top.
- In supplementary figure S4, some network nodes are too crowded to get clear information, such as developmental regulation, is that possible to summarize and compare the distribution of genes on the network NL and Basal-like tumors by using simple boxplot or barplot? At least the network layout should be improved.
- In line 447, what is the correlation for the SETAD2 and cooperation factor EN1 in basal-like cancers?
- In section 2.7, what are the criteria to select the four genes for validation? How many biological replicates are validated for Figure 6?
- In line 574, how does the FTI method convert edge-centric score onto each coherent gene nodes?
- It would be helpful if the author could present the software as an online application.
Reviewer 2 Report
The authors have developed a new Network based Clustering (NetNC) Algorithm to predict the transcription factor target genes using a functional gene interaction network by integrating information from KEGG, STRING and GO databases for Drosophila. After standardizing the algorithm, they applied it on to Twist and Snail ChIP-chip/ChIP-seq data from different groups and predicted their targets. Finally, they mapped Twist and Snail related genes from human breast cancer on to DroFN network and tried to provide new genes and pathways involved in EMT, which are further validated in vitro using invasion assay. The authors need to address the following questions before the paper can be recommended for publication:
- During genome scale map construction for gene functions, was the map constructed between genes or between genes and pathways they are involved in? How many genes had proper annotation on the genome and mapped onto GO and KEGG? Was there no overlapping of genes between the pathways? Did using STRING-DB which considers both direct and indirect interactions between genes not like counting the interactions twice? Why was gene-interaction network integrated and compared with a proteome network for robustness, both serve different purposes. Also, to integrate and use ChIP based data DNA-protein interactions should have be considered instead of protein-protein interactions.
- How do the authors decide the genes as synthetic neutral target genes (SNTG)? Was this done by random sampling of nodes from the whole network and adding them to training data to form the test data? What is gene list size of SNTGs used?
- In Table 1, please include total no. of targets genes identified in the original study in addition to the no. of genes matched in DroFN. In sup Table 4, please mention the no. of genes taken as input (genes identified in the ChIP-on-chip experiment), then compare it with gene which got mapped on to DroFN.
- What method of unsupervised clustering was used for the BrC_2999 with ORTHO_57 and NetNC? How do the results differ if an alternative method is used?
- What was the rational for taking human breast cancer sample gene expression data and mapping onto the DroFN? Why not identify a human functional gene interaction network and perform similar analysis with breast cancer rather than mapping and overlaying the genes between the species?
- In Section 2.7, what is the relationship between four genes selected for the validation with Twist and Snail? Are they regulated by either of them or both? What was the rationale for considering these four genes?
- Further why was Snail taken as EMT inducer for these genes without prior regulatory relation? Provide the expression results of all the genes showing the efficiency of knockdown/induction of respective genes. Also, represent the triplicate data on the graphs and comparisons in a better way.
- Comparing the result of NetNC with already existing software/algorithms doing the similar enrichment analysis is important for benchmarking it with existing softwares.
Round 2
Reviewer 1 Report
Dear authors,
This version of NetNC manuscript has been significantly improved as every major and minor comments addressed. Thanks authors for providing a detailed response letter, in which the authors' answers to all the comments are concrete and supported by new suggested evidences. The method described and corresponding results section are clearer than before, thus some follow-up questions still exist to the following points:
Major 1 and Minor 14: Since the online version is a future plan, then the GitHub command line software is important application for the user. There is no real biological network included in the software so far, it will be helpful to include some model species network (such as DroFN, and mentioned S.aureus networks) with demo real gene sets and refine the software tutorials. In addition, no codes is available for constructing the DroID though the network construction is not the major contribution for NetNC, it will be useful for user to construct more robust network in model species like mouse.
Major 2 and 4: Now it is clear no expression profiles are used in the benchmark dataset. As introduced in the second paragraph in Introduction section, the underlying true gold standards are the differentially expressed genes those are bound by the transcription factor. Then intersection of gene sets between ChIP-chip/seq predicted target genes and microarray/RNA-seq inferred differential gene should be the real benchmark to distinguish neutral binding target genes and functional target genes. However, such a direct benchmark is not used at all as mentioned in Major 4, this is strongly suggested to test within at least one benchmark dataset with paired ChIP and expression data. As a step forward, in Major 2, NEST is slightly worse than NetNC-FBT in Table S3 with simulated network and gene list but it was a slightly unfair since the NEST input gene expression was uniformly sampled. It will be worth comparing NEST and related tools in real dataset with the intersected gene set as benchmark to compare the accuracy and sensitivity. In addition, if no expression available, it might not be able to determine "a significant peak may be correctly assigned to a target gene from a statistical point of view".
Major 5: Authors present a great answer to the question, and suggestion would be to include the survival analysis result into the main Figures.
Minor 7: NFCS score is computed for each node (gene), however, ChIP-seq peak enrichment score is designed for each peak, it is not clear how author performs the peak-to-gene assignment task. Also, within the functional gene set (intersection between differential expression and TF binding genes), the correlation among network score, ChIP-chip/seq gene score, and differential expression would expected to be high.
Minor 10: Figure S5B boxplot have overlaid the x axis, and y axis is recommended to be log transformed.
Minor 12: The selection criteria seems ambiguous, are these four genes within the top 10 genes based on NetFC-FTI FDR rank?
Reviewer 2 Report
The authors have addressed most of my queries.
Author Response
Thanks for your positive endorsement of our work in the review scores and for your time taken on reviewing this manuscript.
Ian Overton, on behalf of all the authors.